# Binding of MAP3773c Protein of *Mycobacterium avium* subsp. *paratuberculosis* in the Mouse Ferroportin1 Coding Region

**DOI:** 10.3390/ijms252312687

**Published:** 2024-11-26

**Authors:** Dulce Liliana Dueñas Mena, José A. Gutiérrez-Pabello, Kaina Quintero Chávez, Mirna Del Carmen Brito-Perea, Dania Melissa Díaz Padilla, Omar Cortez Hernández, José Román Chávez Mendez, Jocelyn Marcela Alcalá Zacarias, Giselle Berenice Vela Sancho, Bertha Landeros Sánchez

**Affiliations:** 1Facultad de Ciencias Químicas e Ingeniería, Universidad Autónoma de Baja California, Tijuana 22390, Mexico; dulce.dueas@uabc.edu.mx (D.L.D.M.); kaina.quintero@uabc.edu.mx (K.Q.C.); dbrito@uabc.edu.mx (M.D.C.B.-P.); dania.diaz@uabc.edu.mx (D.M.D.P.); alcala.jocelyn@uabc.edu.mx (J.M.A.Z.); 2Facultad de Medicina Veterinaria y Zootecnia, Universidad Nacional Autónoma de México, México City 04510, Mexico; jagp@unam.mx (J.A.G.-P.); omarch@fmvz.unam.mx (O.C.H.); 3Facultad de Ciencias de la Salud, Universidad Autónoma de Baja California, Valle de las Palmas, Tijuana 22260, Mexico; roman.chavez@uabc.edu.mx; 4Centro Universitario de Ciencias de la Salud (CUCS), Universidad de Guadalajara, Sierra Mojada 950, Col. Independencia, Guadalajara 44340, Mexico; giselle.vela3080@gmail.com

**Keywords:** MAP3773c, Ferroportin1, transcription factor, transcription regulation

## Abstract

*Mycobacterium avium* subsp. *paratuberculosis* (MAP) is known to cause paratuberculosis. One notable protein, MAP3773c, plays a critical role in iron metabolism as a transcription factor. This study aims to investigate the binding affinity of MAP3773c to the chromatin of the Ferroportin1 (FPN1) gene in murine macrophage J774 A.1. We conducted a sequence alignment to identify potential interaction sites for MAP3773c. Following this, we used in silico analysis to predict binding interactions, complemented by electrophoretic mobility shift assay (EMSA) to confirm in vitro binding of MAP3773c. The map3773c gene was cloned into the pcDNA3.1 vector, with subsequent expression analysis carried out via Western blotting and real-time PCR. Chromatin immunoprecipitation (CHiP) assays were performed on transfected macrophages to confirm binding in the native chromatin context. Our in silico and in vitro analysis indicated that MAP3773c interacts with two binding motifs within the FPN1 coding region. The ChiP results provided additional validation, demonstrating the binding of MAP3773c to the FPN1 chromatin through successful amplification of the associated chromatin fragment via PCR. Our study demonstrated that MAP3773c binds to FPN1 and provides insight into the role of MAP3773c and its effect on host iron transport.

## 1. Introduction

Ferroportin1 (FPN1) is a protein that is so far known as the only one through which iron leaves the cells that contain it [1]. FPN1 [2], called Slc40a1, MTP1, or Ireg1, is found in all tissues where significant iron fluxes are regulated. Several models have been used to study the expression of FPN1. Among the first models used were the J774 mouse macrophages, where metals such as iron, cobalt, copper, zinc, and manganese were used to analyze FPN1 expression. When macrophages were treated with iron, FPN1 expression increased at the level of copper-like mRNA [3]. In the case of cobalt, levels of the FPN1 protein decreased to a significant expression in transcription. In the case of manganese and zinc, the expression of mRNA and proteins did not show significant changes [4]. On the other hand, in mouse models with colitis, whose microbiota influences systemic iron homeostasis and hepcidin expression is altered due to inflammation caused by the intestinal bacterium *Bacteroides fragilis* (*B. fragilis*), it was determined that in macrophages derived from bone marrow, the decrease in the expression of FPN1 was independent of the viability of the bacteria. The decrease in FPN1 induced by *B. fragilis* was functionally crucial, as there was a significant increase in intracellular iron concentrations and a significant reduction in hnRNA (heterogeneous nuclear RNA), and decreased expression occurred at the transcription level. These results also reveal that *B. fragilis* influences the iron management of the inflammatory response of macrophages, modulating the expression of FPN1 [5]. In other models that used bacteria such as *Mycobacterium tuberculosis* (MTB) and *Mycobacterium avium* (MA) to infect mouse macrophages from different tissue populations and in macrophage cell lines from different tissues, expression at the messenger RNA level was found to be different in the various macrophage populations. In another model in peritoneal macrophages and macrophage cell lines, RAW264.7 and AMJ2-C8, mycobacterial infection and IFN-γ stimulation synergistically increased FPN1 mRNA, but not in alveolar macrophages and bone marrow-derived macrophages, where expression was negative. In the case of AM infections, the infected macrophages of the cell lines were RAW264.7 and AMJ2-C8, in which expression increased in synergism with INF gamma. Both bacteria increased the expression of FPN1 as an extract of bacteria and live bacteria [6]. On the other hand, other mycobacteria, such as *Mycobacterium avium *subsp. *paratuberculosis *(MAP), have been used to analyze the behavior of FPN1 expression in J774 mouse macrophages, where expression was analyzed without induction with IFN gamma. Macrophages were treated with MAP extract, with live MAP, with iron, and with MAP and iron together; in this way, it was possible to determine that the expression of FPN1 mRNA in J774 macrophages in those treated with live MAP was slightly increased. However, when treated with iron alone, the concentration increased up to 14 times more URE. When treated with live MAP and iron, the mRNA concentration decreased by up to 70% compared to macrophage treatment with iron [7]. The expression of FPN is thought to be activated at the right time and undoubtedly increases with iron at the transcriptional level. The regulation of expression is found in different stages, from a high level of regulation, i.e., from a unique chromatin structure to post-translational regulation. Post-transcriptional regulation of FPN1 involves the interaction of iron regulatory proteins (IRP1/2) with iron-sensitive element sequences (IREs) within the 5UTR of FPN1 mRNA. On the other hand, transcription factors such as Nrf2 also depended on improving FPN1 transcription in primary macrophages. Prolonged exposure to copper led to a decrease in the abundance of hepcidin and iron-independent FPN1 proteins. The data suggests that cell-type-specific requirements affect the stability of the FPN1 protein after copper loading [8]. Protein–DNA interactions are important in many essential biological processes, such as DNA replication, transcription, and repair. Identifying the amino acid residues involved in DNA-binding sites is critical for understanding the mechanism of these biological activities. In the last decade, numerous computational approaches have been developed to predict protein DNA-binding sites based on protein sequences and structural information. These play an important role in complementing experimental strategies [9]. Our research team, along with Shomaya et al., has identified MAP3773c as a 16.53 kDa protein present in MAP; it has the potential to function as a transcription factor [10] and is a metal regulatory protein that depends on Fe2+ and Zn2+ [11]. This protein contains zinc [12] and regulates cell envelope genes, lipid metabolism of MAP, including MAP3785-MAP3787, MAP2516, and harmful genes of pyruvate metabolism. It also enhances intracellular survival genes such as MAP2309c, MAP2325, and MAP1122 [13]. Studying the function of iron metabolism in the host has been an intriguing area of research. In this study, we aim to explore how the MAP3773c protein interacts with the mouse FPN1 coding regions. To achieve this, we first aligned the sequences of the potential iron boxes where MAP3773c could interact. After identifying these sequences in FPN1, we conducted an in silico molecular docking analysis, followed by an Electrophoretic Mobility Shift Assay (EMSA), Western blotting and a Chromatin Immunoprecipitation (CHiP) assay with a Polymerase Chain Reaction (PCR) end point.

## 2. Results

### 2.1. Alignments 

Figure 1 displays two regions (2482–2503 (GATAATGAGTGTTATCCTTATT) and (ATTTTGACCATCAGAAAGG) 2592–2614) in the coding region of FPN1. These regions were identified based on the similarity of the consensus sequences to which the MAP3773c protein bound. We conducted an alignment with the mouse FPN1, mRNA, (comments in materials and methods), and found similarity in the 2482–2503 region, which we labeled as EBOX due to its similarity to the iron box recognized by *E. coli* [14] FUR-like proteins. The other region with similarity was 2592–2614 in mouse FPN1, which we called MBOX because it resembles regions recognized by the FUR-like proteins of mycobacteria reported by Sala et al. [15]. This allowed us to identify two consensus sequences for molecular coupling experiments.

### 2.2. Docking

Our study aimed to investigate the potential interactions between specific regions of the mouse FPN1 coding sites and the MAP3773c protein of MAP. We used docking simulations to determine the possible interaction of MAP3773c with these regions in FPN1. MAP3773c was analyzed with the MBOX and EBOX coding region of FPN1, using UCSF—CHIMERA molecular visualization programs [16] and AUTDOCK-TOOLS [17]. Coordinates (x,y,z) were utilized for the binding sites to FPN1, MBOX, and EBOX. The size of the grid/box was 80 Å, ensuring coverage of all regions of MBOX and EBOX (Figure 2 and Figure 3). The total area of analysis, based on the provided grid, was 531,441, corresponding to the number of points in dimension X for both MBOX and EBOX, given that there were 80 Å for both cases. It is important to note that the number of atoms for the sequences exceeded the limit accepted by the programs. Therefore, the decision was made to work with the sequence that bound to FPN1, which consisted of nine nucleotides: 5′ ATTTTGACC 3′ and its complement 3′ TAAAACTGG 5′ for MBOX, and 5′GATAATGAG 3′ and its complement 3′CTATTACTC 5′ for EBOX. In the AutoDockTools program, we minimized the twists in the ligand to reduce the possible conformations that the MBOX and EBOX regions could take. This was necessary because even though we are using rigid protein flexible ligand docking, the ligand can adopt many conformations. By restricting the movement of the MBOX and EBOX sequences due to the limitations of rotatable bonds, we aimed to include only the most probable conformations in the model. We covered the ligand entirely to ensure that we did not lose any potential interactions of MAP3773c with the MBOX and EBOX. The files with the pdbqt extension have been generated. Once the study site was selected, the docking process was initiated and executed using the vina.exe program in command prompt [18]; the estimated shift time was 12 h. We present the coupling energies obtained for the interaction of MAP37773c with MBOX and EBOX. The n = 5 shifts were performed to ensure and verify that the procedure was carried out correctly and then to verify the energies. As can be seen, the coupling energies for MBOX-FTPN1 are around −9.2 kcal/mol and −8.7 kcal/mol and for EBOX, from −8.3 to −7.6 Kcal/mol (Figure 2C and Figure 3C). Hence, the study suggests that coupling could be favorable, as it is an exothermic reaction (coupling energies < 0) and could occur spontaneously in a biological system. Some interactions with specific amino acids (Figure 2E and Figure 3D) are also observed, such as ARG45, THR58, GLU57 (enclosed in red), ARG 60, ALA 61, LEU, 67, ARG 71, THR 72, in the case of MBOX. In the case of EBOX, some of the interactions shown are ILE 166, THR 186, GLN 187, ILE 194, TYR 196, LEU 199, and ARG 198. The coupling verification was performed with the HDOCK server [19,20]. The coupling verification involved using the HDOCK server to visualize the interaction between MAP3773c and the MBOX and EBOX encoding region. We utilized the Discovery Studio program [21] to visualize the interaction, as depicted in (Figure 2F and Figure 3F).

### 2.3. The Expression of mRNA and MAP3773c Protein 

In the case of Western blotting, at 6 h of treatment, there was no expression of the MAP3773c protein in any of the treatment groups, and in the same way, at 12 h, only the control of the MAP3773c protein was detected. The experiment was performed in triplicate, and the expression of the treatment of line 6, which corresponds to the cells transfected with pcDNA-*map3773c*, with iron, did not appear; there was only expression in macrophages transfected with pcDNA-*map3773c*, but without iron, at 24 h. The protein was expressed in 12 KDa and not 16.53 KDa, as we expected. Another finding is that in Figure 4, after 48 h, basal proteins can be observed since we used a polyclonal antibody. We also observed a band of approximately 60 KDa, possibly corresponding to a pentamer in treating macrophages transfected with pcDNA-*map3773c* with iron at 48 h since oligomerization is a characteristic of these FUR-type proteins. The 24 and 48 h Western blot protein samples were purified on different dates. They also have different molecular weights since the protein was expressed in pRSET in *E. coli* to be purified, and in this vector, the expression is at 20.53 KDa. Real-time RT-PCR was used to evaluate MAP3773c expression in J774 A.1 mouse macrophages. As shown in (Figure 4), mRNA levels of *map3773c* showed increased expression after 24 h of iron-free treatment and doubled with iron supplementation. However, after 48 h (Figure 4) of iron-free treatment, expression decreased significantly. In contrast, in the presence of iron, expression increased approximately 40-fold compared to the 24 h time point. These findings can potentially open up new avenues of research and deepen our understanding of MAP3773c protein expression.

### 2.4. MAP3773c: Interactions with MBOX and EBOX

Our study investigated the interaction between the MAP3773c protein and the FPN1 coding region in mouse macrophages. We identified two specific regions within the FPN1 encoding sequence, EBOX (2482–2503), GATAATGAGTGTTATCCTTATT and MBOX (2592–2614), ATTTTGACCATCAGAAAGG (Figure 1), and conducted experiments using EMSA. The results showed that the MAP3773c protein interacts with these regions (Figure 2 and Figure 3), displaying the experiment’s outcome. Our findings indicate that the MAP3773c protein can bind to the FPN1 of J774 A.1 mouse macrophages at two coding region sites, even when iron, zinc, and other metals are present, as demonstrated by the addition of EDTA as a metal chelator (Figure 5). This means that MAP3773c does not require metals to bind to these regions of FPN1 in vitro.

### 2.5. MAP3773c Binding in the FPN1 Coding Region

The coding regions of the FPN1 gene were validated by chromatin immunoprecipitation (ChIP) assay. PCR electrophoresis results (Figure 6) indicated that the anti-MAP3773c group and Input shared a band spacing similar to 204 bp, suggesting that MAP3773c may bind with the EBOX and MBOX regions of the mouse FPN1 gene. The negative control with an anti-IgG antibody did not show specific binding because it is an antibody that is explicitly directed against rabbit IgG. In the reaction mixture, there is no rabbit IgG. In contrast, the positive control with anti-RNA Pol II antibody demonstrated specific binding to RNA Pol II in the promoter regions of the genes, for which we amplify a gene, Actin. This binding allowed the capture and amplification of these regions during chromatin immunoprecipitation, both from the MBOX and EBOX boxes, as well as the amplification of the actin gene using oligonucleotides targeting a region of 154 bp. [22]. This showed us that our experimental system was functional.

## 3. Discussion

The FPN1 protein in mammals is responsible for exporting cellular iron into the circulation, making it a key regulator of iron homeostasis [2,23]. Iron plays multiple roles in bacterial growth and pathogenicity and modulates innate immune responses in the host [24].

Our molecular studies, conducted through in silico and in vitro analyses, reveal a potential game-changing finding that MAP impacts iron metabolism within the host, which could influence FPN1 expression. This phenomenon is similar to the effect of other bacteria, such as Bacteroides fragilis, which leads to a decrease in FPN1 expression [5]. Bacteria’s ability to capture iron from host cells to support their growth and carry out various biological processes is a well-known phenomenon, and our research sheds new light on how MTB acquires iron from the surrounding environment [25]. The implications of this finding could be far-reaching, opening up new avenues for understanding and potentially treating diseases related to iron metabolism. This also paves the way for future research, sparking new questions and areas of exploration in the field of iron metabolism and bacterial pathogenicity.

We call these sites EBOX because they resemble the consensus sequence known as the iron box in *E. coli* [14] and MBOX because of their similarity to the mycobacterial iron box obtained through the alignment of mycobacterial sequences [15]. Our findings are the result of a meticulous research process, confirmed by conducting protein–DNA molecular coupling studies and further validated by in vitro analysis using EMSA to evaluate the interaction of MAP3773c in these regions. We analyzed the expression of the MAP3773c protein in vitro in macrophages transfected at 6, 12, 24, and 48 h using Western blotting and real-time PCR only at 24 and 48 h, using Actin expression as a control. The experiment results indicate that the MAP3773c protein was expressed in transfected macrophages 24 h after transfection, appearing as a 12 kDa protein. This suggests that the protein underwent cleavage of its disulfide bonds through a reduction pathway in C94 or C96, likely due to its proximity to the 12 kDa size range.

Notably, the protein was not expressed in macrophages at the 48-h mark, regardless of the presence of iron at the translational level. We analyzed mRNA expression at 24 and 48 h. Protein was expressed at 24 h with and without iron. At 48 h, MAP3773c was expressed with iron only. In the chromatin immunoprecipitation (CHiP) experiments, we found that the MAP3773c protein interacted with the FPN1 coding regions at 24 h in the presence and absence of iron and at 48 h under the same conditions. However, the MAP3773c protein still needed to be expressed at 6 and 12 h, so we opted not to proceed with the experiments at these time points. Our findings indicate that the MAP3773c protein binds to these regions of Ferroportin1 at 24 and 48 h. These results demonstrate that the protein remains expressed for up to 48 h regardless of the presence of iron. These findings align with the EMSA results, which indicate that the protein interacts with these regions irrespective of the presence of metals and the in silico docking results.

Our research has uncovered significant findings, as we have not found documented the participation of FUR-type proteins in the transcription of proteins such as FPN1 in mouse macrophages J774 A.1. Fur and Zur proteins are reported to function as transcription factors in certain organisms, where they form dimers and bind to their consensus DNA sequences. [26]. These findings have been determined only in bacteria, including *Zymomonas mobilis.* This study found that their Fur proteins mediate the expression of stress resistance genes and chromosomal contacts, which are essential for the strain’s adaptation to environmental stimuli [27]. In contrast, regarding FPN1 and its transcription factor regulation, existing documentation supports that Nrf2, a transcription factor commonly found in PC3 cells, upregulates FPN1 expression [28].

The limitations of this study include the inability to work with MAP3773c as a native protein in mycobacteria. It is more practical to use it as a recombinant protein in *E. coli* due to challenges associated with culturing mycobacteria. Another limitation was the potential for the protein to interact with other proteins, which resulted in little or no signal during our attempts to study its interaction with FPN1. Fortunately, we were able to obtain a good signal despite this challenge. We are currently exploring interactions with additional proteins and considering the therapeutic applications of MAP3773c. However, we believe it is too early to provide a definitive analysis, as we still need to understand how MAP3773c influences the expression of FPN1. This will be addressed in our future research.

We will also investigate what other proteins in the FUR family of bacteria may interact in these regions of FPN1. These new findings could lead to innovations in cellular and molecular biology.

## 4. Materials and Methods

### 4.1. Alignments for the Location of Iron Boxes in the Mouse FPN1 Gene

In our study, we conducted a sequence alignment to identify sequences within the coding region of FPN1 that potentially interact with the MAP3773c protein, which is believed to have a transcribing factor function. The FPN1 protein, obtained from GenBank (NC_000067.7: c45965690-45947230 Mus musculus strain C57BL/6J chromosome 1, GRCm39), was the focus of our investigation. We used the FPN1 mRNA sequence from GenBank (BC003438.1) to assess conservation of sequences in the mRNA and their potential impact on protein translation. Our alignments were based on this sequence to identify consensus regions within the gene and to determine any potential interaction between the MAP3773c protein and FPN1 sequence. Additionally, we sought to identify a consensus sequence of mycobacteria that interacts with FUR proteins. To achieve this, we utilized sequences reported by Sala et al. [15] for *M. tuberculosis*, *M. fortuitum*, *M. bovis*, and *M. smegmatis.* Using Geneious 7.1.3 software, we aligned these mycobacterial sequences with the FPN1 sequence to identify similar or identical regions that could potentially interact with MAP3773c. Furthermore, we performed an alignment to identify consensus regions corresponding to the iron box of *Escherichia coli* (*E. coli*), GATAATGATAATCATTATC, as reported by Escolar et al. [14]. Additionally, considering the findings of Shomaya et al. [13], who reported that MAP3773c interacts in this region, we aimed to align the regions based on this antecedent.

### 4.2. Bacterial Strain 

The strain used for protein expression was *E.coli* BL21 (DE3). The procedure used to express and purify the MAP3773c protein was the same procedure already standardized in our laboratory and previously published [12]. 

### 4.3. Docking of the Fur Boxes 

In this study, we utilized a protein model of MAP3773c obtained from homology modeling and crystalline models [12] to perform an in silico coupling analysis. Our aim was to determine the interaction of the protein with the iron boxes or Fur sequences of *E. coli* and mycobacteria. To achieve this, we aligned the protein model with the directions 5′ and 3′ regions of the sequence deposited in the GenBank of MAP3773c and used the Hex 8.0 program and the independent P2Rank method [29,30,31] to predict possible linking sites. This software relies on the Fourier transform and graphics cards to perform the coupling. In the next step, we docked these sequences with the MAP3773c protein, which involves using graphics processors for spatial and electrostatic minimizations. We used Chimera [16,17] to visualize the biomolecule’s possible binding sites of electrostatic and hydrogen interaction. By performing this analysis, we were able to identify potential binding sites between the protein and the target sequences, which could help in further understanding the interaction mechanisms. 

### 4.4. MAP3773c Purification Protein 

The MAP3773c protein was purified in our laboratory with utmost adherence to the standardized methodology, as specified in (12). After purification, the protein was dialyzed in a pH 4.5 and 10 mM sodium acetate solution. For EMSA’s experiments, the protein was precisely used at a concentration of 200 nM. Finally, the protein was quantified. The protein was also used to produce polyclonal antibodies in rabbits. 

### 4.5. EMSA

We conducted an electrophoretic mobility change (EMSA) assay to investigate the interaction of the MAP3773c protein with two potential binding sites in the mouse FPN1 coding region. We follow the EMSA protocol described in [32], with some modifications. In summary, we mixed 44 picomolars/μL of synthesized DNA double-stranded sequences, with a binding buffer (containing HEPES 400 mM, DTT 5 mM and EDTA 2 mM), Ficoll at 20%, MnCl_2_ or ZnCl_2_ at 100 mM and 150 nM, of the protein MAP3773c, 100 picomolar of an unrelated DNA fragment (NRD). We use a fragment of the gene MAP3769c-338 CCTTATTGGGAATCATTTTCATCT (11), with its respective complementary sequence, 0.1 μg poly (dI-dC) and water up to a final volume of 20 μL. Protein-free control reactions were also performed for both cases. For metal-free reactions, we used 2 mM micromolar EDTA [33,34]. The mixtures were incubated for 30 min at room temperature, then loaded into an 8% non-denaturing EMSA acrylamide gel and run at 100 V in a buffer containing Tris 90 nM, boric acid 90 nM and EDTA 10 mM. Finally, we visualized the gels with ethidium bromide using a transilluminator (TFP-M/WL, Upland, CA, USA).

### 4.6. Rabbit Immunization

We performed experiments in accordance with the directive guideline approved by The Animal Care and Use Committee of the Facultad de Medicina Veterinaria y Zootecnia of the Universidad Nacional Autonoma de Mexico, Ciudad de Mexico (project code 158). Polyclonal antibodies against the MAP3773c protein were obtained using proteins in a 0.15 M NaCl solution mixed with aluminum hydroxide Al(OH)_3_ in PBS 1X (1 mL, final volume). A 100 μg protein sample was injected subcutaneously into New Zealand white rabbits. After two weeks, a second immunization was performed, using 100 μg of MAP3773c protein under the same conditions with aluminum hydroxide in PBS 1X. The third immunization was performed intravenously on day 30, using 50 μg of protein in NaCl solution 0.15 M. The fourth immunization was performed the next day, on day 31, under the same conditions as the third, only with 75 μg protein injected intravenously. The fifth immunization was performed on day 32, under the same conditions as the fourth, only 100 μg protein injected intravenously. The serum was obtained after one week and stored at −20 °C until use.

### 4.7. Cloning pCDNA 3.1 and map3773c Gene

The following oligonucleotides were used to amplify the fragment, which contains a recognition sequence for HindIII in the forward primer and for EcoRI in the reverse primer: AAGCTTGTGTCATCGCCCCCTGCGCC (forward) and GATATCCGGTTGTGTGTTTTGGCAAA (reverse). The PCR reactions were purified using the PureLink Quick PCR Kit (Invitrogen, Vilnius, Lithuania), and the resulting fragment was subjected to a double digestion with HindIII-HF and EcoRI-HF enzymes (Biolabs). After digestion, the fragment was purified again using the same kit. The plasmid pCDNA3.1 was also digested with HindIII-HF and EcoRI-HF enzymes and separated on a 1% agarose gel in TAE 1X. The agarose fragment was purified using the PureLink Quick Gel Extraction Kit (Invitrogen, Vilnius, Lithuania). The ligation reaction to fuse the two fragments was performed using T4 DNA ligase (Invitrogen, Carlsbad, CA, USA) at a molar ratio of 1:3 (1 M of pCDNA3.1: 3 M of *map3773c*) in a final volume of 20 μL. 

### 4.8. Sequencing 

To ensure that the gene was accurately linked in the correct direction and size, we performed sequencing on the cloned map3773c fragment located in pcDNA3.1. External sequencing services were employed at the UNAM’s IBT for these procedures. The sequence can be located in the Appendix A.

### 4.9. Macrophage Culture J774 A.1

The murine macrophage cell line, identified as J774 A.1, was procured from the American Type Culture Collection (ATCC, Manassas, VA, USA) and cultured in Dulbecco-modified Eagle medium (DMEM, CORNING, Virginia, USA) supplemented with 10% fetal bovine serum [35] and 1% penicillin-streptomycin. The macrophages were maintained in a monolayer in 25 cm^2^ flasks at 37 °C and 5% CO_2_. The viability of the cell cultures was assessed using the trypan blue exclusion assay. The cells were counted in Newbauer’s chamber and seeded into individual wells for subsequent treatments.

#### 4.9.1. Transfection of Macrophages J774A.1 with pCDNA-map3773c

The transfection experiments were conducted in 6-well plates with a seeding density of 150,000 cells per well and in T75 bottles. The experiments were performed in triplicate. The transfection mixture consisted of plasmid DNA mixed with pCDNA-map3773c with HiperFect Transfection Reagent (Qiagen, Hilden Germany) in DMEM supplemented with 10% FBS and neomycin at a ratio of 37.5 ng plasmid per 150,000 cells in 1 mL of media. For the six-hour trial, after six hours of incubation, macrophages were removed from the transfection mixture, washed with sterile PBS 1X, and the cells were detached using a Cell Scraper, PE Blade (Thermo Fisher, Bohemia, NY, USA). The cells were then collected in sterile 2 mL cryotubes and stored at −70 °C until use. 

The transfection and collection of cells from each treatment were carried out at the corresponding time points, 6, 12, 24, and 48 h.

#### 4.9.2. qRT-PCR

##### Extraction of RNA

We utilized the RNeasy Mini Kit (Qiagen, Hilden Germany), to carry out total RNA extraction according to the manufacturer’s specifications. In brief we processed a maximum of 1 × 10 ^7^ of J774 A.1 macrophage. Finally, 50 μL of nuclease-free water was added to the center of the column, and the sample was centrifuged for 1 min at 8000 rpm before being stored at −70 °C.

#### Reverse Transcription

We performed reverse transcription using the high-capacity cDNA reverse transcription kit from Applied Biosystems. For every 5 μg of total RNA, we combined 1 μL of 100 pmol oligo dT and 1 μL of 0.5 mM, dNTP Mix 10mM, with nuclease-free water to reach a volume of 14.5 μL. Subsequently, we added 4 μL of 5X RT Buffer, 0.5 μL of RNase inhibitor, and 1 μL of Reverse Transcriptase to achieve a final volume of 20 μL for each reaction. The mixture underwent two precise cycles in a BIORAD Thermocycler programmed with the following conditions: 2 min at 25 °C, 10 min at 55 °C, 1 min at 95 °C, and 4 °C for 1 min. Finally, a 5 min cycle at 4 °C was completed. The reaction was then kept at a standby temperature of 4 °C.

#### Real-Time PCR

For the amplification of MAP3773c, we used forward oligonucleotides, GAGCTCGTGTCATCGCCCGCTGGG and reverse AAGCTTTCACGGTTGTGTGTTTTG, respectively, and Actin, and we used the oligonucleotides published in [22]: forward, GGCTGTATTCCCCTCCATCG, reverse, CCAGTTGGTAACAATGCCATGT. We performed two reactions separately; in each one, we put 1 μL oligo forward and reverse prepared at 10 μM, 10 μL of iTaq Universal SYBR^®^ Green Supermix, 5 μL of each cDNA treatment, and then the volume was completed at 20 μL with nuclease-free water. The run was performed on the spot in the Biorad Real-Time PCR System. The equipment is programmed in 1 cycle of 3 min at 94 °C, 40 cycles of 30 s at 94 °C, 40 s at 63 °C, 40 s at 72 °C and finally 10 min at 72 °C.

#### Western Blotting

The Western blotting analysis followed the protocol outlined by Hernandez et al. [12]. Each treatment of macrophage culture contributed forty micrograms of total proteins, which were then separated using a 12% SDS-PAGE gel and transferred onto a nitrocellulose membrane. Subsequently, the nitrocellulose membranes were incubated overnight at 4 °C in 5% milk nonfat dry blotting grade and then with antibodies obtained from rabbits immunized with MAP3773c in milk. These antibodies were diluted 1:16 in 5% nonfat dry milk blotting grade (Apex, Genesee Scientific, El Cajon, CA, USA) in PBS 1X. Subsequently, the membranes underwent three washes with PBS1X-Tween 20, 0.05%. A second anti-rabbit antibody coupled to peroxidase (Sigma-Aldrich, Darmstadt, Germany) was then added and incubated for one hour, followed by another round of washing with PBS 1X-Tween 20, 0.05%. Finally, the membranes were exposed to 3 Amino-9-ethyl carbazole (Sigma-Aldrich, Darmstadt, Germany) and hydrogen peroxide as substrate in 10 mM acetate buffer, pH 5.

#### CHiP

The PierceTM Agarose ChIP Kit (Thermo Scientific, Rockford, IL, USA) was used to carry out chromatin immunoprecipitation (ChIP), following the manufacturer’s instructions. Two million macrophages were transfected and treated with different conditions for 24 and 48 h. Macrophages were transfected with pCDNA-map3773c in the presence and absence of iron nitrile triacetic acid (FeNTA) at a 1:4 molar ratio. Immunoprecipitation was carried out using a polyclonal antibody against MAP3773c, with rabbit IgG serving as a negative control and RNA polymerase-like positive control (Thermo Scientific, Rockford, IL, USA). Precipitated DNAs were identified by PCR using specific primers that detect the two-box (EBOX and MBOX) binding of FPN1 to the mouse gene. 

#### PCR

We conducted a sequence alignment to detect sequences in the coding region of FPN1, where we suspect the MAP3773c protein interacts. To analyze the chromatin immunoprecipitates from different DNA treatments and controls, we conducted PCR to amplify a 204bp FPN1 band using specific oligonucleotides. Our PCR reactions successfully targeted both the EBOX and MBOX in a single reaction, as shown in Figure 6. We used 12.5 μL of EmeraldAmp^®^ GT PCR Master Mix, 1.5 μL of each forward and reverse oligonucleotides. The forward primer for EBOX was 5-GAGCTCCATTACAGAAACAAGTTTTT, and the reverse primer was CCCGGG CCTTAAATAACATACACCTC. For MBOX, the forward primer was 5-GACTTGTCCAAAAGGTTCAT, and the reverse primer was 5-ATACACACATTACTGTAATA. We used only the forward EBOX primer and reverse MBOX primer; the size of the fragments amplified was approximately 204 pb. For Actin, the oligonucleotides used were as follows: forward, GGCTGTATTCCCCTCCATCG, reverse, CCAGTTGGTAACAATGCCATGT, with an expected fragment amplified with 154 pb, 5 μL of DNA, and 4.5 μL of nuclease-free water for the reaction setup. The BioRad T100 Thermal Cycler was utilized with the following cycling conditions: 95 °C for 10 min, 54.5 °C for 30 s, and 72 °C for 45 s. The same reaction and temperature conditions were applied for the actin control. Following amplification, we separated the DNA using 1.2% agarose gel electrophoresis at 100V for 50 min and visualized the DNA bands by staining with ethidium bromide. A 100 bp ladder molecular weight marker was used for reference. 

## Figures and Tables

**Figure 1 ijms-25-12687-f001:**
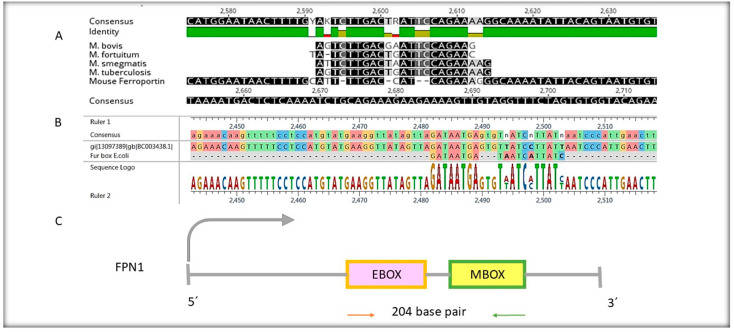
Sequence alignment. (**A**) represents the alignment of mycobacterial sequences with FPN1 of *Mus musculus*. (**B**) represents the alignment of sequences of the iron box of *E. coli* with FPN1 of *Mus musculus*. (**C**) represents the coding sequence of FPN1 and the two internal regions where the two boxes recognized by MAP3773c are located.

**Figure 2 ijms-25-12687-f002:**
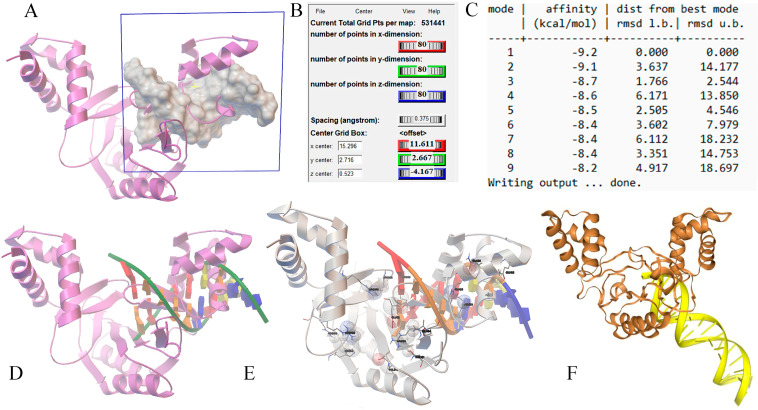
Docking simulations of the MAP3773c protein with the DNA MBOX complexes were conducted to analyze the interaction dynamics. (**A**) spatial analysis was performed to evaluate the interaction sites between the protein and DNA, with specifications detailing the dimensions of interaction in the x, y, and z coordinates. The area of overall analysis (**B**) was quantified, alongside the coupling energies of the complexes (**C**). Key interaction sites between the protein and DNA (**D**) were identified and contact points within the complex were characterized. (**E**) The coupling of MBOX with MAP3773c can be observed at the specified binding site, indicating that they can bind with a binding energy of -9.2 kcal/mol. Interactions with specific amino acids, including ARG45, THR58, and GLU57 (highlighted in red), have also been detected. Additional models of protein–DNA interactions were explored, facilitated by visualizations generated using the Discovery Studio program (**F**).

**Figure 3 ijms-25-12687-f003:**
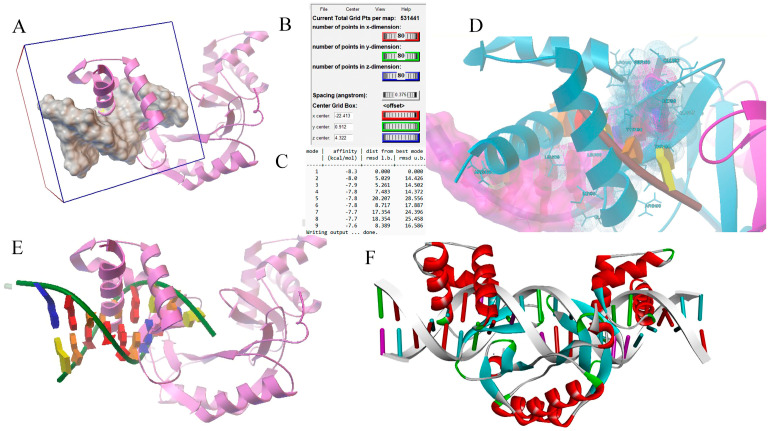
Docking simulations of the MAP3773c protein with the DNA EBOX complexes were conducted to analyze the interaction dynamics. (**A**) A spatial analysis was performed to evaluate the interaction sites between the protein and DNA, with specifications detailing the dimensions of interaction in the x, y, and z coordinates. The area of overall analysis (**B**) was quantified, alongside the coupling energies of the complexes (**C**). Key interaction sites between the protein and DNA (**D**) were identified and contact points within the complex were characterized. (**E**) Alternative interaction visualization format. Additional models of protein–DNA interactions were explored, facilitated by visualizations generated using the Discovery Studio program (**F**).

**Figure 4 ijms-25-12687-f004:**
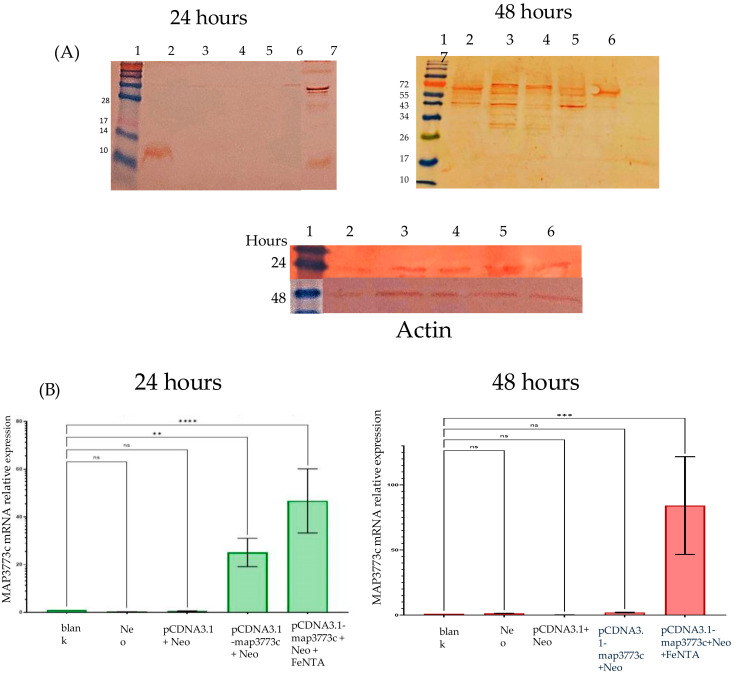
Analysis of Western blotting and real-time PCR of macrophage transfection treatments with pcDNA3.1-map3773c. (**A**) Western blotting using the anti-map3773c antibody against the MAP3773c protein expressed at different time points, 24 h, 48 h and with Anti Actin antibody, 24 and 48 h. The analysis was performed for different treatment groups: line 1, MWM (molecular weight marker); line 2, transfected with pcDNA3.1-map3773c with neomyciny; line 3, not transfected without neomycin; line 4, not transfected with neomycin; line 5, transfected with pcDNA3.1; line 6, transfected with pcDNA3.1-map3773c and with iron 400 nM; and line 7, MAP3773c protein. (**B**) Real-time PCR, 24 h treatment, target, macrophages without transfection and neomycin. Neo, macrophages only with neomycin; pcDNA 3.1+ Neo, macrophages transfected with the vector without the map3773c gene and with neomycin; pcDNA3.1-map3773c +Neo, macrophages transfected with the vector and the map3773c gene and with neomycin; pcDNA3.1–map3773c+Neo+ FeNTA, macrophages transfected with pcDNA-map3773c with neomycin and with iron, but a 48 h of treatment. Actin as internal control. Data are presented as an average ± SD for three independent experiments. ** *p* < 0.01, *** *p* < 0.001 and **** *p* < 0.0001 vs. white; ns, not significant.

**Figure 5 ijms-25-12687-f005:**
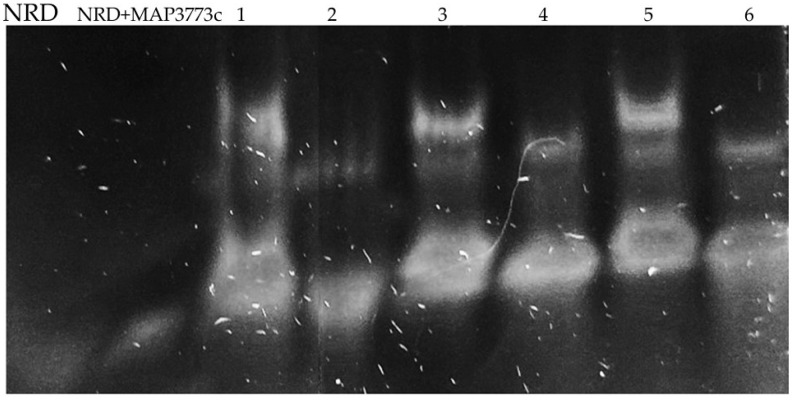
EMSA reaction. The experiments were conducted following the methods and materials described earlier. NRD (MAP3769-338) and NRD with protein (MAP3773c) were tested under four different conditions: (1) region 2482–2503 (EBOX) with two mM EDTA, (2) region 2592–2614 (MBOX) with two mM EDTA, (3) region 2482–2503 (EBOX) with 15 mM MnCl_2_ and (4) region 2592–2614 (MBOX) with 15 mM MnCl_2_. Additionally, two more conditions were tested: (5) region 2482–2503 (EBOX) with 15 mM ZnCl_2_ and (6) region 2592–2614 (MBOX) with 15 mM ZnCl_2_.

**Figure 6 ijms-25-12687-f006:**
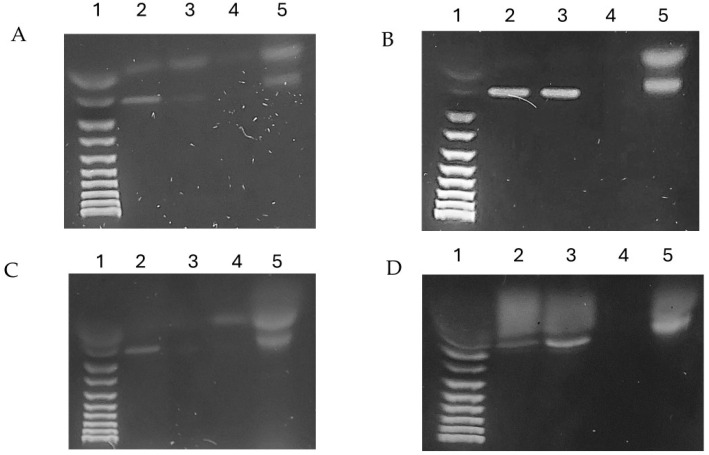
Macrophage transfection CHiP was performed with pcDNA3.1-map3773c at 24 and 48 h for case (**A**): Line 1: Molecular weight marker; 2, CHiP reaction input; 3, CHiP of transfected macrophage DNA pcDNA-map3773c for 24 h; 4, negative control of Anti-lgG rabbit; 5, positive control of chromatin Immunoprecipitation Actin with anti-RNA polymerase antibodies. For case (**B**), the same procedure was followed but the transfection treatments were performed with iron at 24 h. For case (**C**), the same as A, but the treatment duration was 48 h. For case (**D**), the same as B, but the iron treatment duration was 48 h.

## Data Availability

The original contributions presented in the study are included in the article/Appendix A, further inquiries can be directed to the corresponding author.

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
