# Peer review of "Binding of MAP3773c Protein of Mycobacterium avium subsp. paratuberculosis in the Mouse Ferroportin1 Coding Region"

_ijms, 2024, doi:10.3390/ijms252312687_

Round 1
Reviewer 1 Report
Comments and Suggestions for Authors
This paper describes the bioinformatic analysis of proteins that potentially bind MAP3773c through sequence comparison of consensus sequences and molecular docking modelling. The authors go on to show transcription and translation of MAP3773c in transfected cells, but some experiments appear to be missing.
In general the manuscript needs to be checked and fixed for English style writing. There is some Spanish in certain subheadings of the manuscript. Also, the readability and understandability of the text is very poor.
What is the “TAL figure” referring to on line 100 and again on line 189 and line 429?
In Figure 1 the “204 bp” label is off and needs to be corrected. Figure 1 itself is low resolution and difficult read. Especially panels A and B.
Line 101 mentions consensus sequences for binding MAP3773c but does not show them or refer the reader to any sequence in Figure 1.
Figure 2A and 3A – What is Gred Box in panel A? Define this. Figure 2 and 3 legends that describe these figures are not helpful at all.
The entire Section 2.3 should be moved into the Methods section. Also, state that the protein was used to make antibody as well as EMSA studies.
Figure 4 is confusing and hard to interpret. First, I suggest you label the hours beneath each blot rather than make the reader wade through a confusing legend. The Western blot results are not convincing. There should be expression in lane 6, but bands do not co-migrate with the MAP3773c protein in lane 7. Also, blots sizes are different. Panel A could be deleted and just mentioned in the text.
The statement on line 185-186 should be deleted. The data presented shows only binding to FPN1, but not data suggest that this is an unknown mechanism of FPN1 transcriptional regulation.
ESMA was not performed in this study. Only a docking simulation (Figs 2 and 3). ESMA would be run in polyacrylamide gels using a dose-dependent DNA-protein mixture to access binding. The methods in this paper suggest this was done, but the results do not show it. Thus the data do not support the conclusions of MAP3773c binding to or regulating transcription of FPN1.
The ChiP experimental results does not appear to have a negative control. So the conclusion of this experiment is also in doubt.
The discussion paragraph beginning on line 210 way overextends the data. This should be deleted.
In conclusion, the data reported include:
1. A bioinformatic alignment showing consensus EBOX and MBOX sequences in FPN1.
2. A molecular docking simulation suggesting potential binding of the MAP3773c and FPN1 proteins. Again, this is a theoretical bioinformatic approach.
3. Western blot and mRNA transcription expression of MAP3773c transfected macrophages.
4. CHiP experiments, but no ESMA experiment.
Thus there appears to be no solid data showing experimental binding or inhibition of any transcription-translation in iron conditions.
Comments on the Quality of English LanguageThe readability and understandability is very poor at this stage.
Author Response
Cover Letter Addressing Reviewers’ Comments
Dear Reviewers,
We want to express our sincere appreciation for the significant time and effort you dedicated to revising our document. Your insightful suggestions have been invaluable in helping us enhance the overall quality of our article. In the next revised manuscript, we will highlight all changes in yellow. Any modified figures will be enclosed in a black box to make them easily recognizable. We will also address your comments and questions in red.
Once again, thank you for your thoughtful and constructive criticism; it has greatly improved our work.
Reviewer 1
Comments and Suggestions for Authors
This paper describes the bioinformatic analysis of proteins that potentially bind MAP3773c through sequence comparison of consensus sequences and molecular docking modelling. The authors go on to show transcription and translation of MAP3773c in transfected cells, but some experiments appear to be missing.
The article describes a bioinformatic analysis of consensus sequences in Ferroportin 1 of mouse macrophages J774A.1, where MAP3773c binds. To this end, we present a series of experiments to test this effect.
We apologize infinitely for omitting the EMSA experiment results and for the reasons that were sent in that file. As you suggest, we have the corrected article. Thank you infinitely.
In general, the manuscript needs to be checked and fixed for English style writing. There is some Spanish in certain subheadings of the manuscript. Also, the readability and understandability of the text is very poor.
We corrected the writing and made it English. We revised and corrected the paragraphs in Spanish and made them English. If the revised and corrected writing in English needs further corrections, we are willing to have the publisher correct the writing and pay for the service.
What is the “TAL figure” referring to on line 100 and again on line 189 and line 429?
We referenced the figure number, revised the text, and included the corresponding figure number in every line where this legend appears.
In Figure 1 the “204 bp” label is off and needs to be corrected. Figure 1 itself is low resolution and difficult to read. Especially panels A and B.
We will revise the writing and enhance the resolution in panels A and B.
Line 101 mentions consensus sequences for binding MAP3773c but does not show them or refer the reader to any sequence in Figure 1.
We will correct the writing and put the corresponding sequences to which we refer.
Figure 2A and 3A – What is Gred Box in panel A? Define this. Figure 2 and 3 legends that describe these figures are not helpful at all.
The Grid Box is defined and evaluated with, AutoGrid Calculation. Rapid energy evaluation is achieved by precalculating atomic affinity potentials for each atom type in the ligand molecule being docked. In the AutoGrid procedure, the protein is embedded in a three-dimensional grid, and a probe atom is placed at each grid point. The energy of interaction of this single atom with the protein is assigned to the grid point. AutoGrid affinity grids are calculated for each type of atom in the ligand, typically carbon, oxygen, nitrogen, and hydrogen, as well as grids of electrostatic and desolvation potentials. Then, during the AutoDock calculation, the energetics of a particular ligand configuration is evaluated using the values from the grids.", esta descripción viene en la página 4 del manual que le comento.
In Figures 2 and 3, we will revise the legends to enhance clarity and precision in the accompanying descriptions.
The entire Section 2.3 should be moved into the Methods section. Also, state that the protein was used to make antibody as well as EMSA studies.
We will change section 2.3 of the MAP3773c protein purification to the Materials and Methods section and mention that the protein was used to obtain antibodies and for EMSA experiments.
Figure 4 is confusing and hard to interpret. First, I suggest you label the hours beneath each blot rather than make the reader wade through a confusing legend. The Western blot results are not convincing. There should be expression in lane 6, but bands do not co-migrate with the MAP3773c protein in lane 7. Also, blots sizes are different. Panel A could be deleted and just mentioned in the text.
We will mark in Figure 4 the corresponding times of the corresponding treatments in the different sections of Figure 4. We will also make any necessary changes to the legend in Figure 4. It was experimented in triplicate, and the expression of the treatment of line 6 that corresponds to the cells transfected with the vector that includes the map3773c gene and iron did not appear; there was only expression in macrophages transfected with vector and gene, but without iron, the protein was presented in a protein of 12 KDa, and not of 16.53 as we expected its size to be. He probably reduced his disulfide bonds in cysteines 94 and 96. The protein size is 16.53 KDa, expressed in the pcDNA3.1 vector. Another critical issue is that in Figure 4E, a band of approximately 60 KDa can be seen, which corresponds to a tetramer possibly or to a pentamer if the protein was reduced to approximately 12 KDa, which is why the band is not observed at a molecular weight of 16.53 kDa.
It should be noted that the purified protein was expressed in pRSET-A; the protein expressed in this vector is 20.53 KDa, so they do not migrate of the same size. The protein samples in the Western blots were also purified at different times. The one in Figure 4 A, updated 24 hours, had contaminating proteins. We will remove panels A and B; we will only mention them in the text since these treatments have no expression.
The statement on line 185-186 should be deleted. The data presented shows only binding to FPN1, but not data suggest that this is an unknown mechanism of FPN1 transcriptional regulation.
We will remove lines 185-186. For this article, we did not determine if there was any change in the expression of FPN1 at the mRNA and protein levels; these are experiments that we are currently carrying out. However, our results suggest this interaction of MAP3773c in the chromatin of FPN1 of J773 A.1 mouse macrophages.
ESMA was not performed in this study. Only a docking simulation (Figs 2 and 3). EMSA would be run in polyacrylamide gels using a dose-dependent DNA-protein mixture to access binding. The methods in this paper suggest this was done, but the results do not show it. Thus, the data do not support the conclusions of MAP3773c binding to or regulating transcription of FPN1.
Once again, we apologize that, by mistake, we did not present the result of Figure 5, which corresponds to the results of EMSA. We performed as mentioned in the materials and methods section with their respective results responding section. We will put Figure 5 for your review. Therefore, our results validate the results obtained in the in silico and in vitro analysis in CHiP experiments.
The ChiP experimental results does not appear to have a negative control. So, the conclusion of this experiment is also in doubt.
In the development of the CHiP technique, we included a negative control, which consisted of adding to the macrophages transfected with the vector and gene an anti-rabbit IgG antibody that is directed explicitly against an IgG in the reaction, which does not exist in said reaction, so that the antibody does not adhere to any site and at the time of sonicating the chromatin fragments to detach them from the protein and antibody such a complex does not exist, so at the time of amplification by PCR the result is negative. We mentioned in the article that this part of the treatment corresponds to a negative control.
The discussion paragraph beginning on line 210 way overextends the data. This should be deleted.
We will remove that line from the brief.
In conclusion, the data reported include:
- A bioinformatic alignment showing consensus EBOX and MBOX sequences in FPN1.
- A molecular docking simulation suggesting potential binding of the MAP3773c and FPN1 proteins. Again, this is a theoretical bioinformatic approach.
- Western blot and mRNA transcription expression of MAP3773c transfected macrophages.
- CHiP experiments, but no ESMA experiment.
Thus there appears to be no solid data showing experimental binding or inhibition of any transcription-translation in iron conditions.
This study focused on elucidating the interaction of MAP3773c within the FPN1 region through various experimental methodologies. We evaluated the influence of iron availability on the expression levels of MAP3773c, observing a notable enhancement in expression under iron-sufficient conditions. Nevertheless, the assessment of MAP3773c's interaction, irrespective of metal presence, was conducted independently of these variations.
Comments on the Quality of English Language
The readability and understandability are very poor at this stage.
Submission Date
16 October 2024
Date of this review
29 Oct 2024 17:20:01

Reviewer 2 Report
Comments and Suggestions for Authors
This study shows the interaction of MAP3773c protein to Ferroportin1 region in mice and understands the iron homeostasis in MAC subsp. paratuberculosis. This is a well-conducted study that contributes to an area of current clinical importance especially to develop potential therapeutic drugs against mycobacterial infections. Authors has followed right methodology to study the protein/DNA interactions in both in silico and in vitro studies. However, there are a variety of points that should be addressed.
1. Abstract is not clearly written, especially methods. I recommend rewriting this portion with more concise and clearly.
2. Authors didn’t perform any experiments in in vivo conditions but claimed in conclusions and other parts of the manuscript. Please remove if it is not applicable.
3. Why did the authors used E. coli strain for protein expression rather than other mycobacterial strains that will give native proteins.
4. Trim 4.10 section. Since the authors has followed the kit instructions, detailed methodology is not required.
5. In Fig 4, western blots are bit confusing. Untransfected cells are also showing the expression in 3B and 3D. is this basal expression/noise? I would ask authors to provide clear blots with clear indication marks.
6. Revise figure annotations in the footnotes.
7. Study limitations has been discussed. What is the variability of MAP3773C expression and off-target effects while binding. Is there any chance to interact with other iron regulatory proteins? To be explored/validate the therapeutic applications in vivo conditions.
8. The discussion is cumbersome with many repetitions of the same findings. I would try to be condensing it.
9. Authors could have discussed thoroughly on the significance of this study particularly potential therapeutics implications. How this interaction will allow us to target for managing MAC infections. How this could affect MAC pathogenesis.
Minor
10. Please check the punctuations/Grammatical error in the manuscript as they seem to be missing in many places.
11. There are some Spanish words in the manuscript: Eg: line 367, 424, 436, 438.

Author Response
Cover Letter Addressing Reviewers’ Comments
Dear Reviewers,
We want to express our sincere appreciation for the significant time and effort you dedicated to revising our document. Your insightful suggestions have been invaluable in helping us enhance the overall quality of our article. In the next revised manuscript, we will highlight all changes in yellow. Any modified figures will be enclosed in a black box to make them easily recognizable. We will also address your comments and questions in red.
Once again, thank you for your thoughtful and constructive criticism; it has greatly improved our work.
Reviewer 2
Comments and Suggestions for Authors
This study shows the interaction of MAP3773c protein to Ferroportin1 region in mice and understands the iron homeostasis in MAC subsp. paratuberculosis. This is a well-conducted study that contributes to an area of current clinical importance especially to develop potential therapeutic drugs against mycobacterial infections. Authors has followed right methodology to study the protein/DNA interactions in both in silico and in vitro studies. However, there are a variety of points that should be addressed.
- Abstract is not clearly written, especially methods. I recommend rewriting this portion with more concise and clearly.
We will rewrite the abstract clearly and concisely and pay attention to the writing of the methodology.
- Authors didn’t perform any experiments in in vivoconditions but claimed in conclusions and other parts of the manuscript. Please remove if it is not applicable.
We do not perform in vivo experiments so that we will remove the corresponding comments from the manuscript.
- Why did the authors used E. coli strain for protein expression rather than other mycobacterial strains that will give native proteins.
The use of E.coli for the production of the recombinant protein allowed us to work in the laboratory more quickly; we obtained a more significant amount of recombinant protein, which are advantages that allowed us to optimize the conditions of growth of E.coli and purification of the protein than working with some mycobacteria, which takes longer to grow in culture media than in the case of mycobacteria is the main limitation.
- Trim 4.10 section. Since the authors has followed the kit instructions, detailed methodology is not required.
We will cut what is necessary to help the understanding of the technique.
- In Fig 4, western blots are bit confusing. Untransfected cells are also showing the expression in 3B and 3D. is this basal expression/noise? I would ask authors to provide clear blots with clear indication marks.
The spots observed in Figures 3B and 3D represent basal noise. We are using polyclonal antibodies obtained from rabbits. We suspect that the purified protein may contain some residual E. coli proteins, as that is the system we used for protein expression. This could lead to the antibodies recognizing similar macrophage proteins.
Revise figure annotations in the footnotes.
We will assess the texts in the caption.
- Study limitations has been discussed. What is the variability of MAP3773C expression and off-target effects while binding. Is there any chance to interact with other iron regulatory proteins? To be explored/validate the therapeutic applications in vivo conditions.
In this research, we were able to analyze that the expression of MAP3773c in mouse macrophage J774 A.1 was smaller than that expressed using the model in bacteria such as E.coli, where the expression is in a monomer of 20.53 kDa. In macrophages, the protein suffered a reduction in the cysteine regions, so the protein size was approximately 12 KDa; even so, the protein interacted in the coding regions of Ferroportin 1, as we demonstrated. However, there is a possibility that it interacts with other proteins, such as STAT3; for example, since we have done an interaction analysis in silico for other research and the coincidence protein in several transduction signal pathways is STAT3, among others, it is likely that it interacts with other proteins. We are exploring the therapeutic applications of the protein; however, we believe it is premature to issue an analysis without knowing how MAP3773c acts by increasing or decreasing the expression of Ferroportin 1.
- The discussion is cumbersome with many repetitions of the same findings. I would try to be condensing it.
We will review the discussion and remove any unnecessary content.
- Authors could have discussed thoroughly on the significance of this study particularly potential therapeutics implications. How this interaction will allow us to target for managing MAC infections. How this could affect MAC pathogenesis.
The primary objective of this research is to explore novel therapeutic targets that could inform the development of drugs or vaccines for Paratuberculosis. However, as previously discussed in response 7, it is premature to advance this proposal given our current lack of understanding regarding the expression profile of Ferroportin 1 and its potential interactions with MAP3773c or other associated proteins. However, we believe that the interaction of MAP3773c in the FPN1 region may arouse interest in this protein for the control of Paratuberculosis, possibly because MAP3773c would intervene in the expression of FPN1 and this would increase or decrease the development of MAP in the macrophage.
Minor
- Please check the punctuations/Grammatical error in the manuscript as they seem to be missing in many places.
We will examine the punctuation and grammatical errors in the writing.
- There are some Spanish words in the manuscript: Eg: line 367, 424, 436, 438.
We will revise the language according to the suggestions outlined in the letter.

Round 2
Reviewer 1 Report
Comments and Suggestions for Authors
The manuscript is improved with the corrections made.
Comments on the Quality of English LanguageNone specifically, but should be checked carefully by journal production editors.
Reviewer 2 Report
Comments and Suggestions for Authors
1. I would recommend mentioning study limitations. Refer my comments #3 and #6 during first revision.
2. Figure 4: The blots are still unclear. I would ask authors to reframe the blots for better understanding. I couldn’t see figure 4a and 4b annotations in the text.
3. Check the resolution For Figure 5.
